# 1-*O*-alkyl-glycerols from Squid *Berryteuthis magister* Reduce Inflammation and Modify Fatty Acid and Plasmalogen Metabolism in Asthma Associated with Obesity

**DOI:** 10.3390/md21060351

**Published:** 2023-06-07

**Authors:** Yulia Denisenko, Tatyana Novgorodtseva, Marina Antonyuk, Alla Yurenko, Tatyana Gvozdenko, Sergey Kasyanov, Ekaterina Ermolenko, Ruslan Sultanov

**Affiliations:** 1Vladivostok Branch of Far Eastern Scientific Center of Physiology and Pathology of Respiration, Institute of Medical Climatology and Rehabilitative Treatment, 690105 Vladivostok, Russia; curdeal@mail.ru (T.N.); antonyukm@mail.ru (M.A.); yurenko_alla@mail.ru (A.Y.);; 2A.V. Zhirmunsky National Scientific Center of Marine Biology (Far Eastern Branch), Russian Academy of Sciences, 17 Palchevskogo Str., 690041 Vladivostok, Russia; serg724@yandex.ru (S.K.); ecrire_711@mail.ru (E.E.); sultanovruslan90@yandex.ru (R.S.)

**Keywords:** asthma, obesity, obese asthma phenotype, 1-*O*-alkyl-glycerols, plasmalogens

## Abstract

Asthma associated with obesity is considered the most severe phenotype and can be challenging to manage with standard medications. Marine-derived 1-*O*-alkyl-glycerols (AGs), as precursors for plasmalogen synthesis, have high biological activity, making them a promising substance for pharmacology. This study aimed to investigate the effect of AGs from squid *Berryteuthis magister* on lung function, fatty acid and plasmalogen levels, and cytokine and adipokine production in obese patients with asthma. The investigational trial included 19 patients with mild asthma associated with obesity who received 0.4 g of AGs daily for three months in addition to their standard treatment. The effects of AGs were evaluated at one and three months of treatment. The results of the study demonstrated that intake of AGs increased the FEV1 and FEV1/VC ratios, and significantly decreased the ACQ score in 17 of the 19 patients after three months of treatment. The intake of AGs increased concentration of plasmalogen and n–3 PUFA in plasma, and modified leptin/adiponectin production by adipose tissue. The supplementation of AGs decreased the plasma levels of inflammatory cytokines (TNF-α, IL-4, and IL-17a), and oxylipins (TXB2 and LTB4), suggesting an anti-inflammatory property of AGs. In conclusion, 1-*O*-alkyl-glycerols could be a promising dietary supplement for improving pulmonary function and reducing inflammation in obese asthma patients, and a natural source for plasmalogen synthesis. The study highlighted that the beneficial effects of AG consumption can be observed after one month of treatment, with gradual improvement after three months of supplementation.

## 1. Introduction

Asthma is a chronic airway disease with various particular clinical phenotypes and complex pathophysiological mechanisms [1,2]. Nowadays, asthma associated with obesity is classified as a distinct, most severe phenotype which is not easy to control and less responsive to standard asthma medication [3,4,5]. Obesity is an independent risk factor for the pathogenesis of asthma, and it has a significant impact on asthma incidence and manifestations [6]. Chronic systemic inflammation is considered a bridge linking obesity and asthma [7,8]. Studies in immunology have resulted in an extensive evaluation of the inflammatory cells and mediators involved in asthma pathophysiology as well as obesity mechanisms. It is plausible that the hormones responsible for regulating glucose levels and contributing to obesity may have a connection to asthma, both through inflammatory and non-inflammatory pathways [9]. For example, adipose tissue secretes various cytokines and adipokines that can potentially have a detrimental combined impact on the airways [9,10]. The augmented production of certain pro-inflammatory cytokines by adipose tissue in individuals with asthma and obesity may result in clinical implications and affect lung function [11,12,13]. The wide variety of immune mechanisms involved in the development of asthma with obesity is the main reason for the difficulties in controlling and treating this phenotype [14]. Studies have demonstrated that individuals with the obese asthma phenotype exhibit a less favorable response to standard asthma treatments in comparison to asthma patients with a lean body type. Adults with obesity and asthma have reduced responses to asthma treatment courses, resulting in worse disease control, an increased risk of hospitalization, and a decreased quality of life [15]. In the light of mentioned above issues regarding obese asthma therapies, there is a necessity to create a new integrated approach for effective medication and to control the function of the immune and lung systems in obese asthma patients.

Recent evidence suggests that plasmalogens (Pls—a subclass of glycerophospholipids with alkenyl linkages at the sn-1 position of the glycerol moiety, Figure 1) play a crucial role in the physiology and pathophysiology of lung diseases [16,17,18]. 

Over the past two decades, there has been a growing interest in plasmalogens due to their biological roles and links to different ailments [18]. The depletion of plasmalogens, or their reduced amount, is associated with chronic lung disease (asthma, chronic obstructive pulmonary disease (COPD)) [19,20,21], neurodegenerative diseases (Parkinson’s disease, Alzheimer’s disease) [22,23,24], cardiometabolic disease [24,25], and nonalcoholic steatohepatitis [26]. Plasmalogens, as an important component of lung surfactant, may modify the surface tension and surface viscosity of lung surfactants and protect lungs from the aggressive effects of reactive oxygen species [20]. Plasmalogens act as endogenous lipid antioxidants, since the alkenyl linkages are highly susceptible to oxidative damage [27,28]. The participation of plasmalogens in the reduction of the inflammatory response has been established, as well [29,30]. 

Considering the involvement of plasmalogen in the structure and function of the respiratory and immune system, the development of therapeutic strategies aimed at regulating the immune response or modulating lung functions may be an important step in the treatment of asthma. A promising group of compounds capable of influencing the immune function is 1-*O*-alkyl-glycerols (AGs) obtained from marine organisms (Figure 2) [31,32].

1-*O*-alkyl-glycerols are a type of natural ether lipid that can be obtained from neutral lipids with alkyl chains, such as 1-*O*-alkyl-2,3-diacyl-sn-glycerols (DAGE) [33]. On the one hand, AGs are involved in the synthesis of plasmalogens and can be a dietary source for alkenyl glycerophospholipid biosynthesis. The study by Brites P. provided evidence of the beneficial effects of treating a plasmalogen deficiency with the AGs of the Pex7 knockout mouse (mice with a complete deficiency in the biosynthesis of ether-phospholipids) [34]. On the other hand, there is growing evidence showing that dietary marine AGs have a variety of healthy properties such as regulation immune activities, hematopoiesis, reduction in oxidative stress and neuroinflammation, and improvement of the rheological properties of the blood [31,35,36,37]. For example, the most popular source of AGs is shark oil which has been used to boost immunity, modulate the biosynthesis of plasmalogens, and improve the rheological properties of blood [32,38,39]. In a recent study, Sudip Paul showed that supplementation with shark oil leads to significant changes in plasma and circulatory white blood cell lipidomes in overweight or obese participants, such as reductions in total free cholesterol and triglycerides, and modifications of glycerophospholipid molecular species [38]. Alkyl-glycerols isolated from shark liver are able to reduce the risk of developing cardiovascular diseases in obese individuals by regulating the level of vascular endothelial growth factor [40]. In addition, there are various other organisms that contain alkyl-glycerols. Marine creatures (cartilaginous fish, mollusks, sea stars, and others) are considered rich sources of lipids with alkyl or alkenyl linkages, which can be beneficial for drug creation, with an extensive range of pharmacological action [33,37].

Currently, there is a lack of research on the potential benefits of 1-*O*-alkyl-glycerols for individuals with asthma exacerbated by obesity. It is therefore essential to assess the efficacy of AGs in this particular group. Therefore, given the compelling evidence from animal studies, and the paucity of human data on the benefits of AGs in the immune system and lipid metabolism, further exploration of AGs from new marine sources in relation to pulmonary risk factors and lipidomic profiles is warranted.

In the present study, the effect of 1-*O*-alkyl-glycerols from the digestive gland of squid *Berryteuthis magister* on the immune system and lipid metabolism in obese asthma patients has been tested. We examined the influence of AGs on the pulmonary system function, plasma fatty acid and plasmalogen modification, and cytokine and adipokine levels in obese asthma patients.

## 2. Results

### 2.1. Baseline Characteristics of Clinical, Biochemical, and Immune Parameters of Obese Asthma Patients

The study had a total of thirty-five participants who successfully completed it. Patients with asthma (19 individuals: 9 males, 10 females) were subjected to clinical, biochemical, and immunological examinations to evaluate their baseline parameters before beginning the supplementation by 1-*O*-alkyl-glycerols. The investigation involved studying a group of healthy individuals (16 participants: 7 males, 9 females) in order to compare baseline parameters in patients with asthma. The baseline clinical, biochemical, and immune characteristics of patients with asthma and obesity are summarized in Table 1 and Table 2. The average age and body mass index (BMI) of the participants with asthma were 58 years and 34 kg/m^2^, respectively (Table 1). Our findings have revealed that obese asthma patients, despite the basic anti-inflammatory therapy, had elevated levels of inflammatory markers in blood plasma such as leptin, tumor necrosis factor-α (TNF-α), interleukin-2 (IL-2), IL-4, and IL-17a, leukotriene B4 (LTB4), and thromboxane B2 (TXB2). 

We have found that obese asthma patients had high levels of saturated and monoenoic acids such as myristic (14:0), palmitoleic (16:1n−7), hypogeic (16:1n−9), and stearic (18:0) acids in plasma versus the healthy group (Table 2). Plasma concentrations of arachidonic acid (ARA, 20:4n−6) were high in obese asthma patients. At the same time, levels of n−3 polyunsaturated fatty acids (PUFAs), including α-linolenic (ALA, 18:3n−3), eicosapentaenoic (EPA, 20:5n−3), and docosahexaenoic (DHA, 22:6n−3), were decreased. 

The level of total plasmalogens in the blood plasma was evaluated by measuring the content of plasmalogen derivatives, dimethyl acetals (DMA), by gas chromatography–mass spectrometry (GC-MS), and their ratio to fatty acid methyl esters (FAME), which correspond in terms of the number of carbon atoms. The 16:0DMA/16:0FAME ratio was used to indicate the relative content of DMA16:0 to the total content of palmitic acid (16:0) in blood plasma. The 18:0DMA/18:0FAME ratio was used to calculate the content of DMA18:0 to the total content of stearic acid (18:0) in the test sample. Our study has shown that in patients with asthma and obesity, there was a deficiency of plasmalogens in blood plasma compared to a healthy group (Table 2). This deficiency was characterized by a decrease in the level of 16:0DMA, 18:0DMA, and 18:1DMA, and a decline in the ratios of 16:0DMA/16:0FAME and 18:0DMA/18:0FAME.

Based on the data presented, it seems that the standard asthma treatment does not completely alleviate chronic inflammation and cannot address the plasmalogen deficiency. It is crucial to discover a treatment approach that targets all aspects of asthma pathogenesis linked to obesity, including chronic inflammation and imbalances in the composition of fatty acids and plasmalogens. 

### 2.2. 1-O-alkyl-glycerols Improve Lung Function in Obese Asthma Patients

Table 3 displays the pulmonary function of patients with asthma and obesity, as measured by clinical parameters including vital capacity (VC), forced vital capacity (FVC), forced expiratory volume in the first second (FEV1), FEV1/VC ratio, and FEV1/FVC ratio. It has been investigated that AG intake improved pulmonary function after one month and three months of 1-*O*-alkyl-glycerol treatment. For instance, forced vital capacity in the first second (FEV1) increased during treatment and remained unchanged for a more prolonged AG intake. Patients with asthma associated with obesity showed an increase in the ratio of forced vital capacity in the first second to vital lung capacity (FEV1/VC) after consuming AGs for one to three months. This positive trend in pulmonary function improvement indicates that natural ether lipids, such as AGs, can affect the function of the lungs.

### 2.3. 1-O-alkyl-glycerols Reduce ACQ, but Do Not Affect BMI in Obese Asthma Patients

The Asthma Control Questionnaire (ACQ) is a simple questionnaire to assess a patient’s level of asthma control and monitor any changes over time. Higher ACQ scores indicate worse asthma control, and lower ACQ scores correspond with better asthma control [41]. The ACQ is a useful tool for guiding treatment decisions and evaluating the effectiveness of asthma management.

In the present study, the mean ACQ of asthma patients with obesity was 1.34 (Figure 3A). After one month of AG supplementation, ACQ reached clinically significant improvements (>0.5) in seven of the 19 patients, and in 17 of the 19 patients after three months of treatment. Overall, the study has shown that AGs had a positive effect on the ACQ score of obese asthma patients. 

Body mass index (BMI) did not change during the treatment (Figure 3B). It is worth noting that in order to minimize the number of variables that could affect the outcome, patients maintained a standard diet during the trials of AG efficacy. As a result, we did not anticipate observing substantial weight loss as part of this observation. By having patients adhere to their standard diet, we can be more confident that any changes in the patient’s condition are due to the drug being tested, rather than other factors, such as diet changes.

### 2.4. Effects of 1-O-alkyl-glycerol Supplementation on Plasmalogen Level in Obese Asthma Patients

As mentioned above, obese asthma patients had lower levels of plasmalogens in plasma compared to the healthy group. This lack is identified by a reduction in the levels of 16:0DMA, 18:0DMA, and 18:1DMA, as well as a decrease in the ratios of 16:0DMA/16:0FAME and 18:0DMA/18:0FAME. It has been shown that after 1 month of treatment with AGs, levels of 16:0DMA and 18:1DMA increased in obese asthma patients (Figure 4A,C). The ratio of 16:0DMA/16:0FAME and 18:0DMA/18:0FAME increased, as well (Figure 4E). A similar trend is observed in the third month of AG supplementation. The level of 16:0DMA and 18:1DMA increased compared to healthy volunteers and compared to results obtained before treatment (Figure 4A,C). 

These findings demonstrated that the consumption of AGs through diet can serve as a reliable source of plasmalogen synthesis, thereby correcting its deficiency in asthma and obesity. 

### 2.5. Effects of 1-O-alkyl-glycerol Supplementation on Plasma Fatty Acid and Oxylipin Level in Obese Asthma Patients 

Plasma fatty acid modification allows us to assess not only the intensity of lipid metabolism, but also the activity of immune processes and the ability of cells to synthesize inflammatory, anti-inflammatory, and pro-resolving lipid mediators. It has been known that obesity has a great impact on lipid metabolism, thereby impairing the activity of immune cells and the synthesis of inflammatory lipid mediators such as oxylipins [42]. Indeed, the results of the study of the composition of fatty acids in blood plasma and the level of oxylipins in patients with asthma show a disturbance in lipid metabolism. This was indicated by increased levels of saturated and arachidonic fatty acids and a decreased concentration of n-3 PUFA in the blood plasma of obese asthma patients (Table 2).

There were slight differences seen in the levels of saturated fatty acids with AG supplementation for one month and for three months (Table 4). The amount of 14:0 and 18:0 gradually decreased from one to three months of AG consumption. In contrast, the levels of 18:3n−3, 20:5n−3, and 22:6n−3 were significantly increased after AG supplementation. Three months of treatment became more effective in terms of n−3 PUFA modification. 

A decrease in the proportion of saturated fatty acids circulating in the blood in people with asthma and obesity after AG treatment may suggest an improvement of metabolic processes in the liver and a shift in the priority of synthesis from saturated to unsaturated fatty acids. 

It has been shown that intake of AGs reduced LTB4 and TXB2 generation by immune cells. Obese asthma patients had a lower level of LTB4 (Figure 5A) and TXB2 (Figure 5B) after one month of AG treatment. The level of inflammatory oxylipins continued to reduce during the three months of AG intake (Figure 5). Our results suggest that AGs have very strong anti-inflammatory effects as well as an ability to modify fatty acid metabolism. 

### 2.6. 1-O-alkyl-glycerols Regulate Adipocyte Adiponectin and Leptin Generation in Obese Asthma Patients

Adiponectin and leptin are among the vast variety of adipokines produced by adipose tissue and have opposite effects. Leptin resistance and hyperleptinemia in obesity enhance inflammation and a wide range of pathogenic mechanisms. In contrast, adiponectin has an anti-inflammatory effect, and in obesity the level of adiponectin usually decreases [43].

Our findings suggest that obese asthma patients had elevated levels of leptin in their blood plasma, indicating systemic chronic inflammation and an inflammatory response of the adipose tissue (Table 1). Supplementation with AGs did not have a statistically significant effect on leptin levels after one month of treatment (Figure 6A). However, after three months of AG intake, the leptin levels decreased compared to the baseline. This suggests that long-term supplementation with AGs may have a beneficial effect on leptin levels in obese asthma patients by reducing the inflammation of adipose tissue. 

The baseline level of adiponectin in obese asthma patients was reduced (Table 1), which is in line with existing studies confirming that in obesity and asthma, adiponectin synthesis is inhibited in response to chronic inflammation [11,44]. Additional dietary intake of AGs has increased the level of adiponectin in plasma for one month and three months of treatment (Figure 6B). There were no differences in adiponectin between one month and three months of AG treatment. 

These findings demonstrate that the intake of AGs can modify leptin/adiponectin production by adipose tissue, improving the balance between pro-inflammatory and anti-inflammatory adipokines.

### 2.7. 1-O-alkyl-glycerols Modulate Immune System Function and Reduce Chronic Inflammation

It has been known that the obese asthma phenotype has a Th2 low profile with the predominant neutrophil type of inflammation. This type of chronic inflammation is characterized by a high generation of inflammatory cytokines such as IL-6, TNF-α, IL-17, IL-1β, and IFN-γ. Higher serum IL-6 levels in adults with asthma have been found to correlate with increased body weight, decreased lung function, and an elevated risk of exacerbations [3,7,10,45]. Our results have confirmed that patients with asthma and obesity have increased levels of circulating inflammatory cytokines such as TNF-α, IL-2, IL-4, and IL-17a despite the basic anti-inflammatory therapy (Table 1). 

1-*O*-alkyl-glycerol supplementation for one and three months has had a positive effect on immune system activity. After one month of AG treatment, levels of TNF-α, IL-4, and IL-17a were decreased (Figure 7B,E,F). In contrast, there were no differences in IL-2, IL-6, IL-10, and IFN-γ generation after one month of AG treatment. Long-term intake of AGs has reduced the levels of IL-2, IL-4, IL-17a, and TNF-α, showing an anti-inflammatory effect (Figure 7A,B,E,F). Additionally, supplementation with AGs resulted in a statistically significant decrease in plasma IL-17A, suggesting a reduction in neutrophilic inflammation, a hallmark of the obese asthma phenotype (Figure 7E).

The results showed that AGs had a significant suppressing effect on both leptin and cytokines. However, there were no significant differences in INF-γ, IL-6, and IL-10 (as shown in Figure 7). Our findings also revealed that treatment with AGs regulates low-grade chronic inflammation by reducing the synthesis of inflammatory mediators.

## 3. Discussion

Over several decades, there has been an increased interest in studying the pharmacological properties of 1-*O*-alkyl-glycerols (AGs) from marine organisms [31,32,38,39,46,47,48]. 1-*O*-alkyl-glycerols exhibit diverse biological activities, including anti-inflammatory, antioxidant, and antitumor effects [27,32,36,46,47,48]. There are a number of studies that showed the adaptogenic and hematopoietic properties of AGs [31,32]. The effects of AGs have been investigated in the context of various diseases, including cancer, liver disease, and neurodegenerative diseases [22,25,26,27,31,36,37,49]. However, their impact on the obese asthma phenotype remains unexplored. To address this knowledge gap, the current study aims to investigate the effects of AGs from the squid *Berryteuthis magister* on immune cell and adipocyte activities, as well as fatty acid and plasmalogen metabolism in patients with asthma and obesity.

This open-label investigational study evaluated the clinical and immunological effects of AGs (0.4 g/day) in asthma-obese patients. The selection of the dosage was established based on previous clinical studies [38,39,46]. Each dose was administered for three months. This study displayed an enhancement of treatment efficacy of obese asthma patients using AG supplementation with basic glucocorticoid and β2-agonist therapy. It has been known that one of the main problems in the management of patients with asthma and obesity is resistance to glucocorticoids and difficulties controlling asthma symptoms [2,15]. Gaining weight can significantly affect lung function by causing restriction due to increased fat accumulation around the chest wall and abdomen. This can lead to a decrease in lung capacity and expiratory reserve volume, resulting in ventilation/perfusion mismatching [50]. In addition to these effects, obesity can also lead to greater bronchial hyperresponsiveness, which means that the airways may become more sensitive to certain triggers such as allergens, pollutants, or respiratory infections. This can increase the likelihood of developing asthma or exacerbating existing asthma. On top of this, obesity is associated with a chronic state of low-grade inflammation, which can exacerbate asthma symptoms [7,8].

Our investigation demonstrated that consuming AGs on a daily basis for a period of one to three months resulted in an improvement in pulmonary function, as evidenced by increased forced expiratory volume in the first second (FEV1) and FEV1 to vital capacity ratio (FEV1/VC). These results of pulmonary function improvement associated with AG treatment suggest that natural ether lipids can have a positive impact on lung function. The Asthma Control Questionnaire score (ACQ) reduction observed in asthmatic and obese patients who received AGs also provides evidence of the beneficial effect of these marine lipids on the lung function and overall health of participants, despite the lack of weight loss. Although weight loss has been shown to improve clinical and immunological parameters in asthma [51], persuading patients to modify their dietary habits and restrict calorie intake can be challenging. Incorporating AGs into the standard treatment protocol could enhance the condition of such patients who are, for different reasons, unable to lose weight. 

The statistically and clinically significant improvement in asthma control measured by the ACQ, found in our study, presumably is the result of the influence of AG consumption on plasmalogen and fatty acid metabolism. Herein, using lipidomics we have revealed an increase in 16:0 dimethyl acetals (DMA) and 18:1DMA concentration in plasma in obese asthma patients under AG treatment. After AG supplementation, we also observed an increase 16:0DMA/16:0 fatty acid methyl esters (FAME) and 18:0DMA/18:0FAME ratio, indicating the rise of lipids with alkyl linkages relative to acyl lipids. Moreover, the results of the study of alkyl lipids in blood plasma showed that the long-term intake of AGs had a greater impact on the elevation of the plasma plasmalogen level of asthmatic patients. 

Plasmalogens (Pls) are an important component of lung surfactant, and their modification may be impact lung function [20]. The data found by Mario Rüdiger showed that preterm infants with an increased percentage of plasmalogens and polyunsaturated fatty acids in the pulmonary surfactant system are protected against the development of bronchopulmonary dysplasia [21]. Sordillo et al. first identified plasmalogens as possible mediators of the age-related changes in lung function of individuals suffering from asthma [19]. The lack of plasmalogens in lung tissue mediates the toxic effect of atmospheric ozone and smoking on lungs [52]. The alkenyl linkage of plasmalogen is highly susceptible to pro-oxidants, thereby preventing oxidative damage to cell membrane components. As a precursor of plasmalogens, AGs behave as an antioxidant and help to eliminate reactive oxygen species [27,53]. In addition, oxidative stress induces a decrease in plasmalogen levels [27,54]. Presumably, intake of AGs can help alleviate plasmalogen deficiency in the body, including in the lungs, which improves the structure and function of the surfactant. This improvement may have a positive impact on the overall functioning of the lungs.

Furthermore, we have shown the beneficial influence of AGs on the plasma fatty acid composition in individuals with obesity-related asthma. The levels of 14:0 and 18:0 gradually decreased from one to three months of AG consumption. Conversely, the concentrations of n−3 PUFAs (18:3n−3, 20:5n−3, and 22:6n−3) significantly increased following AG supplementation. The three-month treatment period was more effective in modifying n-3 PUFA levels.

Plasmalogens play a vital role in liver fatty acid metabolism, regulating the key biochemical pathways of lipid synthesis [16,26]. The administration of the plasmalogen precursor, AGs, prevents hepatic steatosis and non-alcoholic steatohepatitis by activating fatty acid oxidation through PPARα. Furthermore, treatment with AGs restored liver plasmalogen levels and increased liver DHA-plasmalogen concentrations [26]. Plasmalogens have an impact on the homeostasis of cholesterol and n−3 and n−6 polyunsaturated fatty acids (PUFAs), particularly arachidonic acid (ARA, 20:4n−6), and docosahexaenoic (DHA, 22:6n−3). They also serve as precursors for platelet-activating factor (PAF) and lysoplasmalogens. As a result, the effects of plasmalogens can be seen through the various biologically active substances they produce and the associated signaling pathways [31,55,56]. 

Taking the above-mentioned investigations together with our data, we can conclude that AGs are a beneficial source of endogenous plasmalogens, and their administration can improve fatty acid metabolism as well. Moreover, our study indicated the ability of marine AGs to inhibit the generation of inflammatory oxylipins such as as leukotriene B4 (LTB4), and thromboxane B2 (TXB2). Considering that LTB4 and TXB2 are produced from arachidonic acid via the lipoxygenases (LOX) and cyclooxygenase-1 (COX-1) pathways [57], the decrease in their levels following treatment suggests that AGs are capable of regulating the synthesis of oxylipins, most likely by impacting the metabolism of PUFAs.

The anti-inflammatory role of lipids with an alkyl bond may be due to the fact that plasmalogens block the activation of nuclear factor (NF)-kB and mitogen-activated protein kinases (MAPKs) such as JNK and p38 MAPK, thereby attenuating the nuclear translocation of the NF-kB subunit and inhibiting inflammatory signaling pathways [58]. A reduction in cellular Pls enhanced the endocytosis of Toll-like receptor 4 (TLR4) and triggered the activation of caspase-3, ultimately leading to an increase in the expression of pro-inflammatory cytokines [35]. Remarkably, when administered to mice with an Alzheimer’s disease model, Pls were found to significantly reduce TLR4 endocytosis, providing further evidence for the anti-inflammatory effects of 1-*O*-alkyl-glycerols through an additional molecular mechanism.

In support of the anti-inflammatory effects of AGs, our study has shown that a dietary intake of AGs was associated with low circulating concentrations of pro-inflammatory cytokines, e.g., interleukin-2 (IL-2), IL-4, Il-17a, and tumor necrosis factor-α (TNF-α), and a decreased level of adipocyte produced leptin. Since the obese asthma phenotype is distinguished by persistent, low-grade inflammation that stems not only from airway inflammation but also from heightened activity in adipose tissue (adipocytes produce a number of cytokines and adipokines, which may have a synergistic adverse effect on the airways), a high AG diet might provide comprehensive anti-inflammatory properties. 

Numerous studies have demonstrated that neutrophilic inflammation is a relevant mechanism of the obese asthma phenotype. The abundant neutrophil level is associated with the presence of higher levels of IL-17A, a cytokine involved in neutrophil recruitment of the airways [7,59]. The obese asthma phenotype is also related to the increased production of pro-inflammatory cytokines—TNF-α and IL-1β in the lung [12], and IL-6 in serum [9]. In our study, we identified the neutrophilic type of inflammation in patients with asthma and obesity and proved the contribution of AGs to the improvement of immune parameters. Natural AGs decreased IL-17A and TNF-α generation, thereby reducing the inflammatory burden of the disease. In addition, long-term supplementation with AGs had a more beneficial effect on levels of inflammatory mediators in obese asthma patients, although further research is needed to confirm this and to understand the underlying mechanisms. 

Thus, the consumption of natural AGs derived from marine sources, in combination with basic medication for patients with mild asthma and obesity, can lead to a reduction in chronic inflammation by decreasing the generation of pro-inflammatory cytokines and oxylipins. The observed changes, along with the notable improvements in clinically significant indicators of asthma and inflammation, present a compelling rationale for conducting larger trials to investigate the effects of AG supplementation on lung diseases.

To summarize, our study has demonstrated several beneficial effects of supplementing with AGs in individuals with asthma and obesity. These effects include improvements in lung function, fatty acid and plasmalogen metabolism, and immune activity. Dietary supplements containing AGs were observed to modulate plasma plasmalogen levels and effectively reduce chronic inflammation in individuals with asthma associated with obesity. The mechanisms underlying these effects are complex and require further investigation, particularly in the context of pulmonary diseases and the obese asthma phenotype. Considerable effects achieved by the intake of AGs are the elimination of plasmalogen deficiency and improvement in disease control, as indicated by decreased ACQ. Furthermore, the study revealed that these beneficial effects can be observed after one month of alkyl-glycerol supplementation and continue to improve after three months of treatment.

It is important to acknowledge that our study has certain limitations, including the absence of a robust placebo control or cross-over design, as well as a limited number of participants with asthma and obesity. We only focused on results after one and three months of AG supplementation. It is valuable to assess 1-*O*-alkyl-glycerols’ long-term impact and investigate the duration of these active compounds’ anti-inflammatory effect post-discontinuation. In spite of these limitations, the study certainly adds new knowledge about the impact of AGs on lung function, the immune system, and plasmalogen and fatty acid synthesis in obese asthma patients and shows the anti-inflammatory and lipid-metabolism-improving properties of marine AGs. 

## 4. Materials and Methods

### 4.1. Study Population and Sample Selection 

This was a parallel, controlled clinical trial. Participants completed written informed consent before the trial and were allocated into two groups: the AG and control (healthy) groups. Patients of the AG group received, alongside the standard prescription treatment, a dietary supplement AGs for 3 months. All subjects gave their informed consent for inclusion before they participated in the study. The study was conducted in accordance with the Declaration of Helsinki, approved by the Ethics Committee of the Institute of Medical Climatology and Rehabilitative Treatment, Vladivostok, Russia (protocol No. 14/04-12-2018, approved on 4 December 2018). This clinical study included 19 patients of both sexes (9 male, 10 female) with mild asthma associated with obesity, recruited at the Clinical Department of the Institute of Medical Climatology and Rehabilitative Treatment, Russia. Asthma was diagnosed based on the guidance of the Global Strategy for Asthma Management and Prevention [2]. The comparison group were 16 healthy control subjects ≥25 years of both sexes (7 male, 9 female). Criteria for inclusion in the control group: volunteers who fit the definition of “conditionally healthy”—the absence of respiratory pathology, chronic infectious and non-infectious pathology, with a negative allergic history, and not burdened by COPD, asthma, and other allergic and chronic diseases.

The patients with asthma were selected to participate in the study if their asthma was controlled or partially controlled according to GINA criteria. The study participants continued to follow their usual diet while participating in the study, which means that we did not impose any dietary restrictions or interventions. This helped us to minimize any potential confounding factors and increase the external validity of the study results.

The ACQ-5 test (Asthma Control Questionnaire) was used to determine the level of asthma management; the scores ranged 0.75–1.5, which implied the presence of partially controlled asthma. Outcomes included Asthma Control Questionnaire (ACQ) score, the immune state, cytokine and adipokine levels, blood plasma fatty acid and plasmalogen concentrations, and plasma eicosanoid level.

The external respiration function was assessed using the Master Screen Body apparatus (Care Fusion, Höchberg, Germany). The analyzed parameters were vital capacity (VC), forced vital capacity (FVC), forced expiratory volume in 1 sec (FEV1), and FEV1/VC and FEV1/FVC ratio. The reversibility of airway obstruction was evaluated by bronchodilation test with inhaled salbutamol (400 μg). According to current guidelines from the European Respiratory Society (ERS) and the American Thoracic Society (ATS), the absolute and relative changes in FEV1, also known as the bronchodilation ratio, provide a reliable evaluation of the reversibility of airway obstruction. A positive result in the bronchodilation test is achieved when the bronchodilation coefficient is over 12% and the absolute increase in FEV1 is more than 200 mL [40].

Obesity was established according to the WHO recommendations [60,61]. Body mass index (BMI, weight/height, kg/m^2^) was calculated to diagnose overweight or obesity. Based on the WHO classification, normal body weight was defined as BMI 18.9–24.9 kg/m^2^, overweight as BMI 25–29.9 kg/m^2^, obesity Class I was diagnosed as BMI 30–34.9 kg/m^2^, obesity Class II as BMI 35–39.9 kg/m^2^, and obesity Class III as BMI ≥ 40 kg/m^2^.

Inclusion criteria for the study included: male and female persons aged 18–60 years old; asthma from mild to moderate severity, controlled and partially controlled course; overweight and obesity of class 1 or 2. Exclusion criteria included: severe asthma, secondary obesity; class 3 obesity, acute illness, chronic diseases (such as diabetes, metabolic syndrome, thyroid diseases, pancreas insufficiency), pregnancy and lactation, cancer, intake of n-3 PUFA or 1-*O*-alkyl-glycerol nutritional supplements, or missing consent. All subjects gave written informed consent prior to the survey. 

Overnight fasting blood samples were obtained early in the morning from healthy volunteers and patients with asthma. 

### 4.2. Treatment and Supplementation

All asthma patients were provided with basic therapy with a fixed combination of a low-dose inhaled glucocorticoid (budesonide: 160 micrograms/day) and a long-acting β2-agonist (formoterol 4,5 micrograms/day) (steps 1 and 2 of GINA-2019).

The dietary supplement «Incodamarine» («BIONIKA» LLC, Tyumen, Russia, State Registration No. AM.01.01.01.003.R.000866.12.22) includes natural AGs extracted from the digestive gland lipids of the squid *Berryteuthis magister*. «Incodamarine» contains AGs: chimyl alcohol (91.07%) and batyl alcohol (7.38%) (see Table 5). «Incodamarine» is presented in the form of capsules with dry powder (0.2 g) of a mixture of AGs.

Patients, in addition to the standard prescription treatment (described above), received a dietary supplement of AGs for 3 months, 1 capsule 2 times a day after meals, which is equivalent to 0.4 g of pure AGs. The dose selection was determined through previous clinical studies on the use of AGs and the recommended effective doses of AG-containing supplements found in the instructions. The effects of AG supplementation were evaluated in obese asthma patients at one and three months of treatment (Figure 8). 

### 4.3. Analysis of Total Fatty Acids and Dimethyl Acetals in Plasma

Blood from participants was drawn into VACUETTE^®^ EDTA tubes with EDTA and samples were centrifuged with 3000 g at 40 C for 10 min. The plasma was collected and stored at −80 C before analysis. The gas chromatography–mass spectrometry GC-MS (Shimadzu, Kyoto, Japan) method was used to analyze plasma total fatty acids and plasmalogens [62,63,64]. The extraction of lipids from plasma was performed using the Bligh and Dyer method [62]. To prepare the methyl esters of fatty acids (FAMEs) and dimethyl acetals (DMAs) from plasma lipids, we followed the procedure outlined by Carreau and Dubacq [63]. The α,β-unsaturated ether in the plasmalogen molecule was converted into a DMA of the corresponding aldehyde during transesterification, and the relative amount of plasmalogen was thus reflected in the ratio between 18:0 aldehyde and stearic acid as well as by the ratio between 16:0 aldehyde and palmitic acid [65]. A gas chromatography analysis of FAME was conducted on GC-MS 2010 (Shimadzu, Kyoto, Japan). DMA identification was carried out by comparison of retention times with standards of 16:0 DMA and 18:0 DMA (Sigma-Aldrich, Merck, Massachusetts, USA). GC-MS (TQ-8040, Shimadzu, Kyoto, Japan) was utilized to confirm the FAME and DMA structures. 

### 4.4. Analysis of Eicosanoid Concentrations

Plasma TXB2 and LTB4 concentrations were determined using enzyme immunoassay kits from Enzo Life Sciences (Farmingdale, NY, USA). Columns with reversed-phase C18 (Minicolumns for Sample Preparation, USA) for extraction were pre-washed with 10 mL of ethanol >95% and then with 10 mL of deionized water. Then a sample was applied and the column was washed with 10 mL of water, then 10 mL of 15% ethanol and finally 10 mL of hexane. The sample was eluted from the column by adding 10 mL of ethyl acetate and dried on the IKA MPV10 basic (GmbH) rotary evaporator. Then the ELISA method was performed. The 96-well plates were read on an Automatic Immunoassay analyzer (Evolis Twin-Plus, BIO-RAD, Hercules, CA, USA). TXB2 and LTB4 concentrations were calculated from a standard curve, according to the kit protocol.

### 4.5. Cytokine and Adipokine Level Analysis

The levels of IL-2, IL-4, IL-6, IL-10, TNF-α, IFN-γ, and IL-17A cytokines in the serum were measured using flow cytometry with a BD FACSCantoII cytometer and the “BD Cytometric Bead Array—Human Th1/Th2/Th17 Kit” (USA). The data was processed using “FCAP Array 3.0” software from BD (Version 3.0, San Jose, CA, USA). Commercial ELISAs were used to determine serum concentrations of adiponektine (Human Adiponectin ELISA) and leptin (DRG ELISA) according to manufacturer’s instructions. The plates were read on an Automatic Immunoassay analyzer (Evolis Twin-Plus, BIO-RAD). 

### 4.6. Statistical Analysis

The statistical analysis and graphical illustration of assays were carried out using the GraphPadPrism 9.5.0 software (GraphPad Software, Inc., New York City, NY, USA). The data underwent a one-way ANOVA test, followed by a Tukey’s multiple comparison test. The results were presented as the mean ± standard error of the mean (SEM) or as median, and upper and lower quartiles. A *p*-value less than 0.05 was considered statistically significant. 

## Figures and Tables

**Figure 1 marinedrugs-21-00351-f001:**
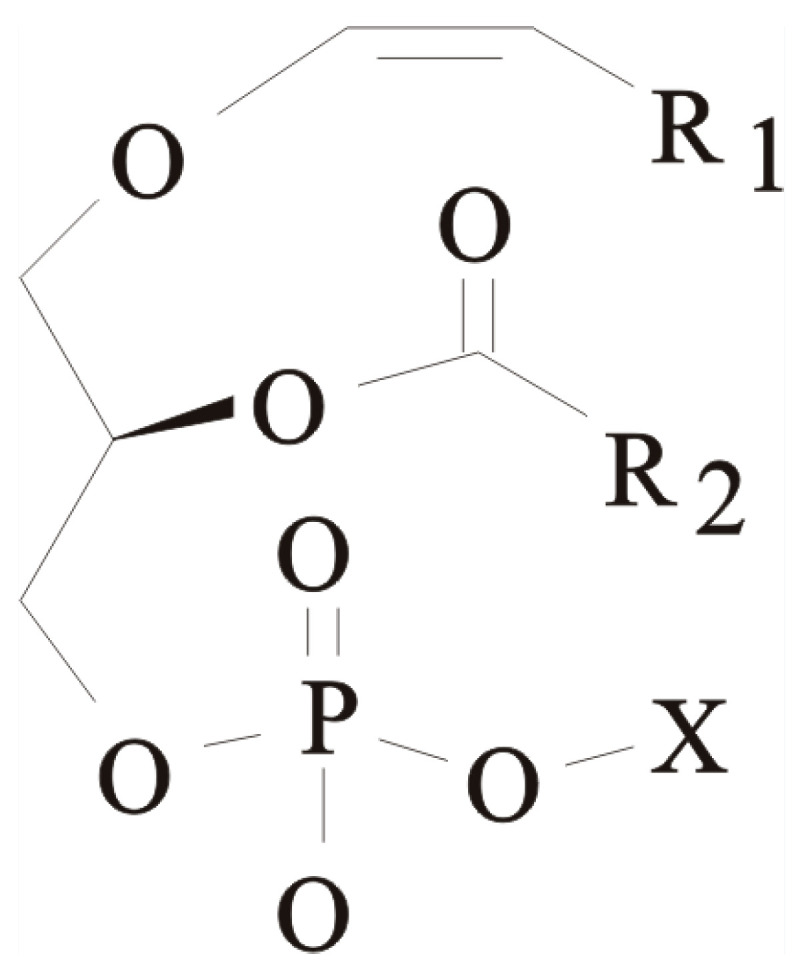
Chemical structure of 1-*O*-(1Z-alkenyl)-2-acyl-sn-phospholipid (plasmalogens), where R1 is a hydrocarbon fragment of fatty alcohols and aldehydes and R2 is a hydrocarbon fragment of fatty acids. X—ethanolamine, choline or serine.

**Figure 2 marinedrugs-21-00351-f002:**
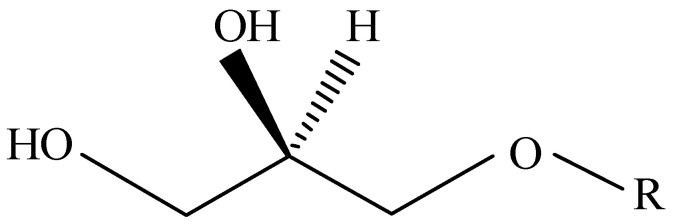
The chemical structure of 1-*O*-alkyl-glycerol where R—12:0, 14:0, 16:0, 18:0, 16:1, or 18:1 alkyl chain.

**Figure 3 marinedrugs-21-00351-f003:**
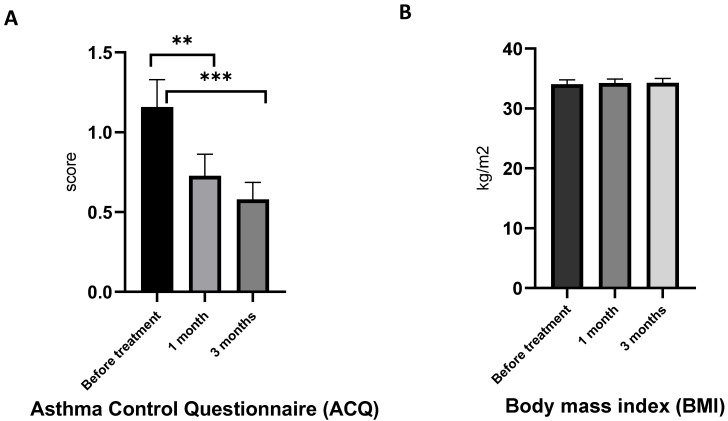
Effect of AGs on the Asthma Control Questionnaire (ACQ) and Body mass index (BMI) in obese asthma patients after one month and three months of treatment. (**A**) ACQ score under AG treatment; (**B**) BMI in obese asthma patients after one and three months of 1-*O*-alkyl-glycerol supplementation. Data are presented as mean ± SEM, ** *p* < 0.01, *** *p* < 0.001 (one-way ANOVA, Tukey’s post-test study).

**Figure 4 marinedrugs-21-00351-f004:**
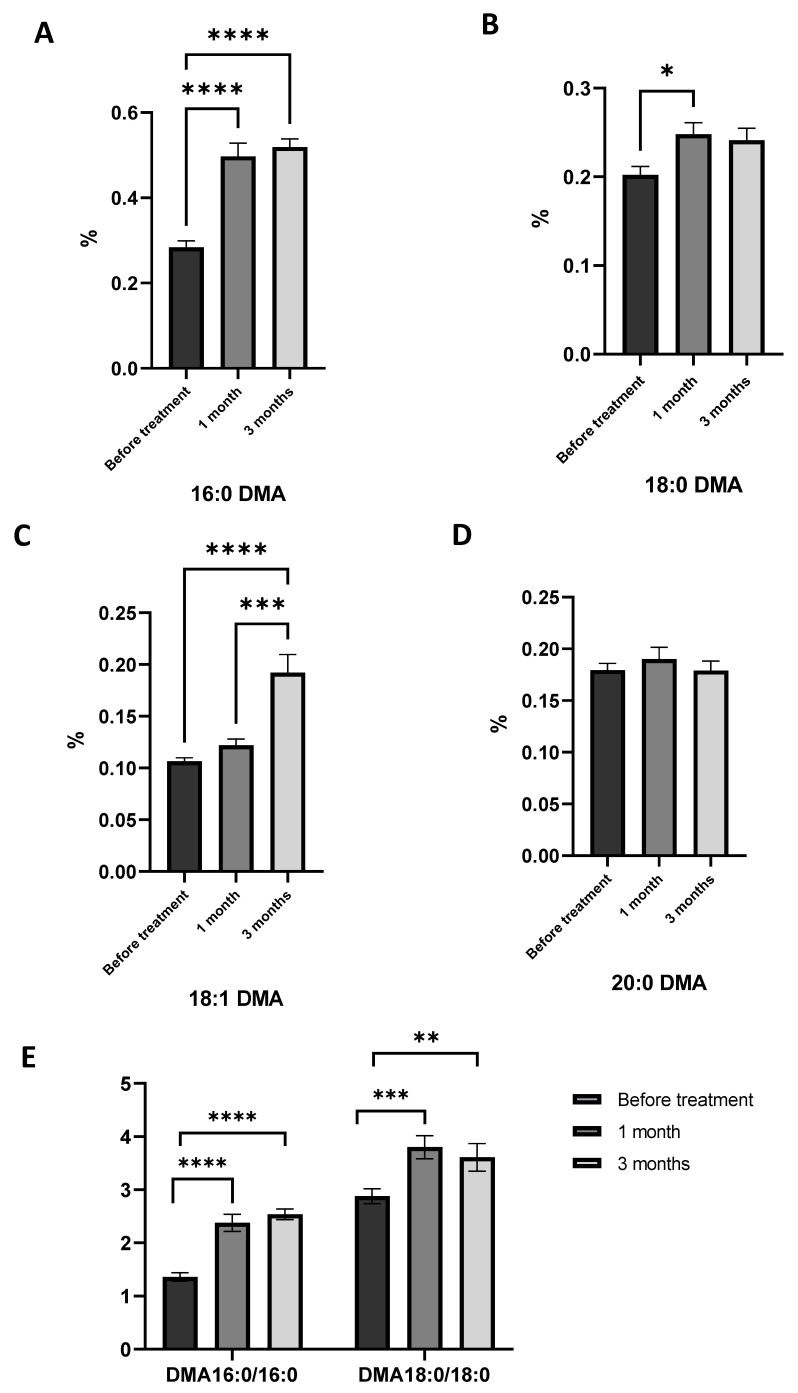
Impact of 1-*O*-alkyl-glycerols on plasma levels of plasmalogens in obese asthma patients. Plasmalogens were analyzed using GC/MS and represented as % from total fatty acids. DMA—dimethyl acetals, FAME—fatty acid methyl esters. (**A**) Level of 16:0 DMA in the blood plasma of obese asthma patients before and after one and three months of 1-*O*-alkyl-glycerol supplementation. (**B**) Level of 18:0 DMA in the blood plasma of obese asthma patients before and after one and three months of 1-*O*-alkyl-glycerol supplementation. (**C**) Level of 18:1 DMA in the plasma of obese asthma patients before and after one and three months of 1-*O*-alkyl-glycerol supplementation. (**D**) Level of 20:0 DMA in the blood plasma of obese asthma patients before and after one and three months of 1-*O*-alkyl-glycerol supplementation. (**E**) 16:0DMA/16:0 ratio and 18:0DMA/18:0 ratio under AG treatment. Data are presented as mean ± SEM, * *p* < 0.05, ** *p* < 0.01, *** *p* < 0.001, and **** *p* < 0.0001 (one-way ANOVA, Tukey’s post-test study).

**Figure 5 marinedrugs-21-00351-f005:**
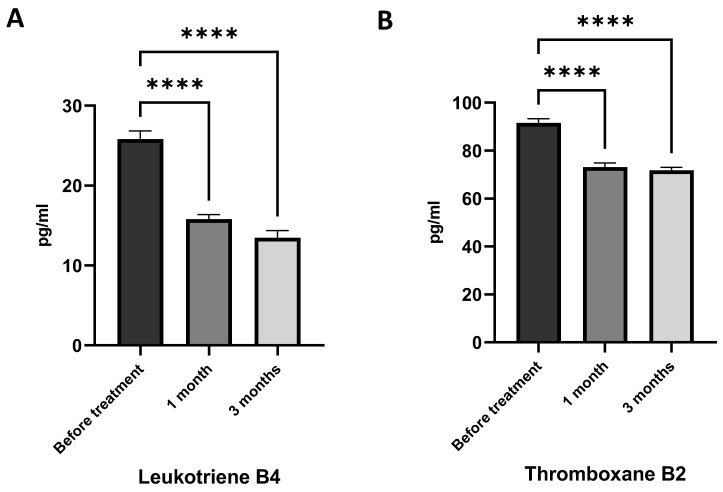
Impact of 1-*O*-alkyl-glycerols on the plasma levels of inflammatory oxylipins in obese asthma patients. Plasma leukotriene B4 (LTB4) and thromboxane B2 (TXB2) concentrations were determined by ELISA. LTB4 and TXB2 levels were calculated from a standard curve, according to the kit protocol, and represented as pg/mL. (**A**) LTB4 level in plasma of obese asthma patients before and after one and three months of 1-*O*-alkyl-glycerol supplementation. (**B**) TXB2 level in plasma of obese asthma patients before and after one and three months of 1-*O*-alkyl-glycerol supplementation. Data are presented as mean ± SEM, **** *p* < 0.0001 (one-way ANOVA, Tukey’s post-test study).

**Figure 6 marinedrugs-21-00351-f006:**
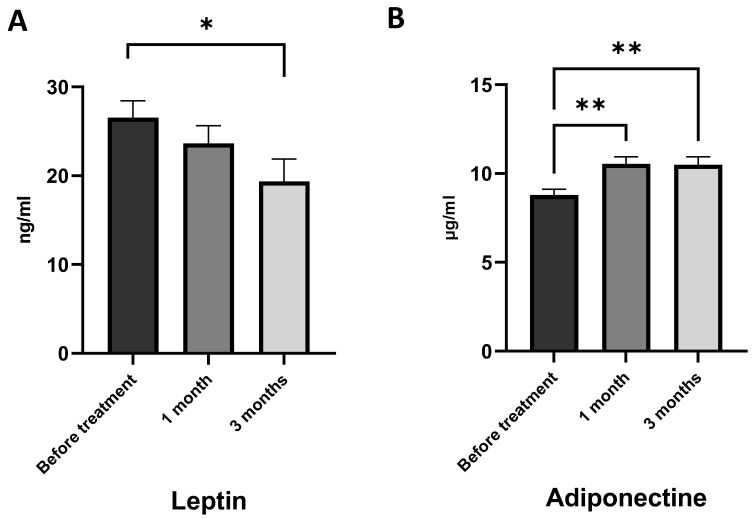
Impact of 1-*O*-alkyl-glycerols on the leptin and adiponectin generation of obese asthma patients. Plasma leptin and adiponectin concentrations were determined by ELISA, and represented as ng/mL for leptin and µg/mL for adiponectin. (**A**) Plasma level of the leptin of obese asthma patients before and after one and three months of 1-*O*-alkyl-glycerol supplementation. (**B**) Plasma level of the adiponectin of obese asthma patients before and after one and three months of 1-*O*-alkyl-glycerol supplementation. Data are presented as mean ± SEM. * *p* < 0.05, ** *p* < 0.01 (one-way ANOVA, Tukey’s post-test study).

**Figure 7 marinedrugs-21-00351-f007:**
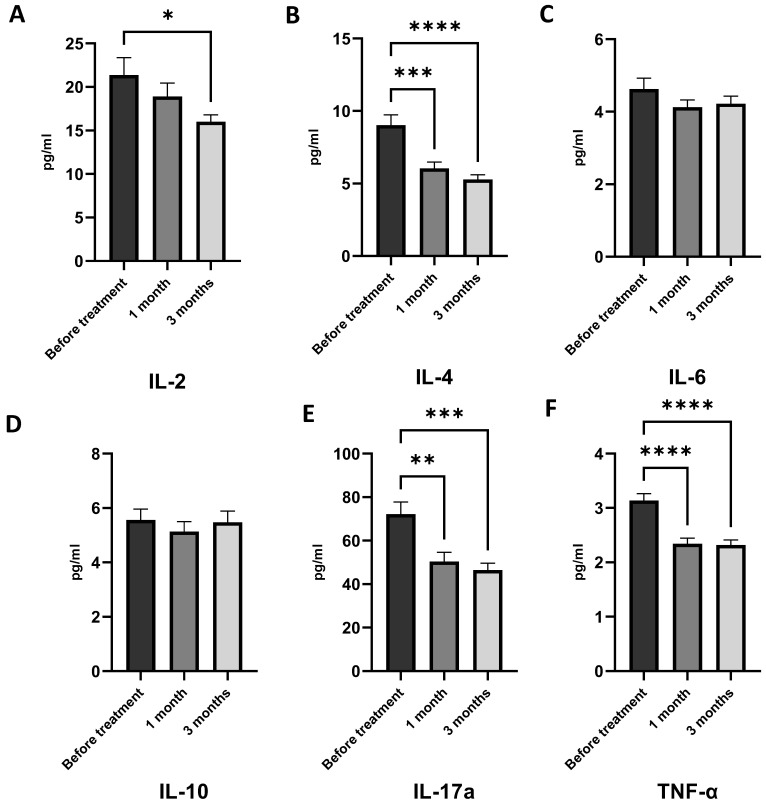
Effect of 1-*O*-alkyl-glycerols on the cytokine levels of plasma in obese asthma patients. Plasma cytokine concentrations were determined by flow cytometry and represented as pg/mL. (**A**) Plasma level of the IL-2 of obese asthma patients before and after one and three months of 1-*O*-alkyl-glycerol supplementation. (**B**) Plasma level of the IL-4 of obese asthma patients before and after one and three months of 1-*O*-alkyl-glycerol supplementation. (**C**) Plasma level of the IL-6 of obese asthma patients before and after one and three months of 1-*O*-alkyl-glycerol supplementation. (**D**) Plasma level of the IL-10 of obese asthma patients before and after one and three months of 1-*O*-alkyl-glycerol supplementation. (**E**) Plasma level of the IL-17a of obese asthma patients before and after one and three months of 1-*O*-alkyl-glycerol supplementation. (**F**) Plasma level of the TNF-α of obese asthma patients before and after one and three months of 1-*O*-alkyl-glycerol supplementation. (**G**) Plasma level of the INF-γ of obese asthma patients before and after one and three months of 1-*O*-alkyl-glycerol supplementation. Data are presented as mean ± SEM. * *p* < 0.05, ** *p* < 0.01, *** *p* < 0.01, **** *p* < 0.0001 (one-way ANOVA, Tukey’s post-test study).

**Figure 8 marinedrugs-21-00351-f008:**
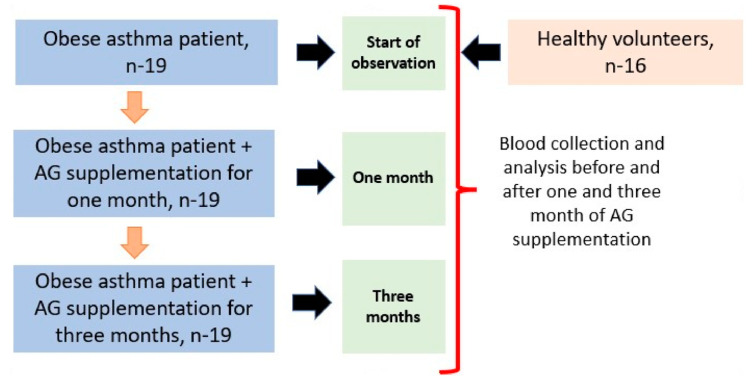
Study design for AG supplementation in obese asthma patients. Individuals were enrolled in the research and requested to attend an initial assessment. During the visit, a medical evaluation was conducted to determine their suitability. Those who were deemed eligible were called back and assigned to take AGs for a period of three months. Fasting blood samples were collected from each participant at the initial screening, after one month of AG supplementation, and after three months of AG treatment. All participants successfully completed the treatment regimen.

**Table 1 marinedrugs-21-00351-t001:** Comparison data of clinical and immune parameters between healthy volunteers and obese asthma patients.

Characteristics	Healthy	Asthma with Obesity
Age	45.88 ± 8.54	58.31 ± 7.01
Body mass index (BMI), kg/m^2^	23.50 ± 0.72	34.06 ± 0.72 ***
Asthma Control Questionnaire (ACQ)	-	1.08 ± 0.15
Adiponectin, mg/mL	11.94 ± 0.60	9.32 ± 0.38 ****
Leptin, ng/mL	2.03 ± 0.21	26.53 ± 1.89 ****
Interleukin-2 (IL-2), pg/mL	13.30 ± 0.73	21.36 ± 2.00 **
Interleukin-4 (IL-4), pg/mL	3.39 ± 0.19	9.02 ± 0.70 ***
Interleukin-6 (IL-6), pg/mL	4.49 ± 0.28	4.62 ± 0.30
Interleukin-10 (IL-10), pg/mL	5.29 ± 0.19	5.56 ± 0.40
Tumor necrosis factor-α (TNF-α), pg/mL	2.31 ± 0.09	3.13 ± 0.13 ***
Interferon gamma (IFN-γ), pg/mL	28.79 ± 2.57	28.93 ± 1.43
Interleukin-17a (IL-17a), pg/mL	36.05 ± 1.29	72.10 ± 5.62 ***
Thromboxane B2 (TXB2), pg/mL	63.18 ± 2.50	91.56 ± 1.75 ****
Leukotriene B4 (LTB4), pg/mL	12.06 ± 0.61	25.81 ± 1.02 ****

Data are presented as mean ± SEM, with significance indicated by asterisks. An asterisk indicates a comparison to the healthy group. ** *p* < 0.01; *** *p* < 0.001; **** *p* < 0.0001.

**Table 2 marinedrugs-21-00351-t002:** Comparison of the plasma fatty acid levels of healthy participants and obese asthma patients (% of total fatty acids).

Fatty Acids, %	Healthy	Asthma with Obesity
14:0	0.51 ± 0.02	1.04 ± 0. 05 ***
15:0	0.18 ± 0.01	0.19 ± 0.008
16:0	20.27 ± 0.37	21.76 ± 0.33
16:1n−9	0.39 ± 0.009	0.47 ± 0.01 ***
16:1n−7	1.46 ± 0.07	2.02 ± 0.09 ***
18:0	6.32 ± 0.13	7.05 ± 0.11 ***
18:1n−9	15.80 ± 0.41	16.71 ± 0.25
18:1n−7	1.52 ± 0.05	1.55 ± 0.02
18:2n−6	37.17 ± 0.76	36.01 ± 0.59
18:3n−6	0.32 ± 0.02	0.36 ± 0.02
18:3n−3	0.53 ± 0.01	0.47 ± 0.01 *
20:3n−6	1.16 ± 0.06	1.17 ± 0.04
20:4n−6	5.14 ± 0.12	5.47 ± 0.17 *
20:5n−3	0.88 ± 0.07	0.78 ± 0.09 *
22:4n−6	0.18 ± 0.01	0.12 ± 0.004 ***
22:5n−3	0.46 ± 0.03	0.41 ± 0.029
22:6n−3	2.32 ± 0.15	1.50 ± 0.04 ***
16:0DMA	0.41 ± 0.02	0,28 ± 0.01 **
18:0DMA	0.27 ± 0.01	0.20 ± 0.00 **
18:1DMA	0.14 ± 0.006	0.10 ± 0.00
20:0DMA	0.16 ± 0.007	0.17 ± 0.00
DMA16:0/16:0FAME	1.97 ± 0.14	1.31 ± 0.07 ***
DMA18:0/18:0FAME	4.14 ± 0.25	2.87 ± 0.14 ***

Data are presented as mean ± SEM, with significance indicated by asterisks. An asterisk indicates a comparison to the healthy group. * *p* < 0.05; ** *p* < 0.01; *** *p* < 0.001. DMA—dimethyl acetals, FAME—fatty acid methyl esters.

**Table 3 marinedrugs-21-00351-t003:** Effect of 1-*O*-alkyl-glycerols on lung function tests in obese asthma patients.

Lung Function Parameters, %	Before Treatment	One Month	Three Months
VC	108.7 (98.1;118.8)	110.2 (102.3;118.8)	110.9 (105.3;119.4)
FVC	107.2 (98.2;116.0)	109.78 (101.7;116.0)	109.5 (101.2;117.8)
FEV1	105.2 (99.7; 112.4)	108.6 (102.8;113.3) *p* < 0.001	109.0 (102.9;114.1) *p* < 0.001
FEV1/VC	87.0 (76.2;98.3)	90.2 (77.9;101.3) *p* < 0.05	91.0 (78.3;102.5) *p* < 0.05
FEV1/FVC	83.1 (71.5; 91.3)	86.7 (80.6;91.3)	86.9 (80.8;92.2)

The data are presented as median, and upper and lower quartiles. The table represents the results of the lung function of obese asthma patients (19 participants) before AG treatment, after one month of AG supplementation, and after three months of AG supplementation. VC—vital capacity, FVC—forced vital capacity, FEV1—forced expiratory volume in the first second.

**Table 4 marinedrugs-21-00351-t004:** Effect of 1-*O*-alkyl-glycerol supplementation on plasma fatty acid level in obese asthma patients (% of total fatty acids).

	Asthma with Obesity		
Fatty Acids, %	Before Treatment	One Month	Three Months
14:0	1.04 ± 0.05	0.94 ± 0.06	0.89 ± 0.04
15:0	0.19 ± 0.008	0.19 ± 0.007	0.18 ± 0.00
16:0	21.76 ± 0.33	20.97 ± 0.38	20.57 ± 0.34
16:1n−9	0.47 ± 0.01	0.41 ± 0.01 *	0.42 ± 0.01
16:1n−7	2.02 ± 0.09	2.00 ± 0.09	2.36 ± 0.13
18:0	7.05 ± 0.11	6.63 ± 0.15 *	6.81 ± 0.14
18:1n−9	16.71 ± 0.25	16.32 ± 0.24	16.48 ± 0.3
18:1n−7	1.55 ± 0.02	1.53 ± 0.03	1.54 ± 0.03
18:2n−6	36.01 ± 0.59	34.54 ± 1.00	36.78 ± 0.74
18:3n−6	0.36 ± 0.02	0.33 ± 0.02	0.34 ± 0.02
18:3n−3	0.47 ± 0.01	0.51 ± 0.03 **	0.52 ± 0.03 **
20:3n−6	1.17 ± 0.04	1.23 ± 0.03	1.25 ± 0.03
20:4n−6	5.47 ± 0.17	5.04 ± 0.18	4.91 ± 0.2 *
20:5n−3	0.78 ± 0.09	1.01 ± 0.11 **	0.91 ± 0.06 **
22:4n−6	0.12 ± 0.004	0.13 ± 0.006	0.13 ± 0.00
22:5n−3	0.41 ± 0.029	0.46 ± 0.025	0.45 ± 0.02
22:6n−3	1.50 ± 0.04	2.16 ± 0.13 ***	2.08 ± 0.08 ***

Data are presented as mean ± SEM, with significance indicated by asterisks. An asterisk indicates a comparison to obese asthma patients before treatment with AGs. * *p* < 0.05; ** *p* < 0.01; *** *p* < 0.001.

**Table 5 marinedrugs-21-00351-t005:** Composition of 1-*O*-alkyl-glycerols in «Incodamarine».

Alkyl Chain ^a^	Content, % ^b^	Trivial Name
14:0	1.55 ± 0.20	
16:0	91.07 ± 0.21	chimyl alcohol
18:0	7.38 ± 0.57	batyl alcohol
Ʃsat	100 ± 0.00	

^a^ is showing the chain length and double bond of alkyl chain in AGs. ^b^ is showing the percentage (%) of component to total components (mean value ± standard deviation (*n* = 5).

## Data Availability

The data presented in this study are available upon request from the corresponding author.

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
