# Peer review of "1-O-alkyl-glycerols from Squid Berryteuthis magister Reduce Inflammation and Modify Fatty Acid and Plasmalogen Metabolism in Asthma Associated with Obesity"

_marinedrugs, 2023, doi:10.3390/md21060351_

Round 1

Reviewer 1 Report

This study of Yulia Denisenko revealed the protective effect of Marine-derived 1-O-alkyl-glycerols (AGs) Asthma Associated with Obesity. This study showed that ingestion of 0.8g daily ameliorate lung function, reduce inflammation, decrease leptin; and adiponectin generation…

My recommendations are:

1.      The plagiarism rate is 27%, it must be reduced to 20%

2.      The number of patients used is too few 19 patients?

3.      It is necessary to add the number of experimentation ethic

4.      Why the choice of the 0.8g/day dose? also the dose should be based on body weight

Reviewer 2 Report

This a well written paper that aimed to investigate the effect of AGs from squid Berryteuthis magister 15 on lung function, fatty acid and plasmalogen levels, cytokine and adipokine production in obese 16 patients with asthma

Figure C is not understandable, Please include the bar charts

Table 2. Please remove the column healthy, it counduses the analysis

dividing for categories and time is not appropriate. you can compare status and time in 2 different plots. Please apply this strategy

It is not fruitful looking diseases and time in the same x axis. It does not make sense at all.

Please discus
